# WARPED LATENT SPACES AND TRAVERSAL FOR CHEMICAL DEEP GENERATIVE MODELS

## ABSTRACT

We propose a generative framework for interpretable and property-aware molecular design by learning warped subspaces within the latent space of a chemical variational autoencoder (VAE) trained on a sequential representation of small molecules. Instead of directly regularizing latent coordinates, our approach works by creating low dimensional subspaces that are smoothly warped to align with molecular property variation using a novel alignment loss. This warping provides a flexible mechanism to capture non-linear structure in property–latent relationships while retaining interpretability. This framework enables property optimisation and traversal within a low-dimensional subspace, where directions correspond to meaningful variations in molecular properties and decode back into valid molecules in the original space. We evaluate the method on various tasks related to conditional molecular generation on standard benchmarks used in literature like QM9, ZINC250K and the Pubchem drug datasets demonstrating strong generative quality, validity, uniqueness and novelty alongside a more controllable approach.

## 1 INTRODUCTION

Deep generative models (DGMs) have become an integral part of the discovery process in modern science and engineering (Bilodeau et al., 2022; Jørgensen et al., 2018; Ravanbakhsh et al., 2017; Lopez et al., 2020). This work is about leveraging generative latent variable models for small molecule generation — a predominant modality for modern drugs. A critical bottleneck in the drug discovery process is the identification of new molecules from an overwhelming search space ($\approx 10^{60}$) (Polishchuk et al., 2013). It is interesting to note that small-molecule drugs discovered as recently as in the early 2000s were using traditional phenotypic assays, i.e., by directly observing their effects on disease physiology (Swinney and Anthony, 2011). In contrast, de novo design implies designing a potentially novel chemical compound with an optimal property profile for downstream in vitro testing and synthesizability. Generative models offer a viable solution to this inverse design problem by fundamentally learning a probability distribution of molecular data which can be used not only for unconditional generation of molecules but also for property constrained generation by exploration of a so-called *chemical space* through gradient-based techniques (Gómez-Bombarelli et al., 2018).

A generative model in the small-molecule context essentially features such an open-ended chemical latent space learned by embedding discrete molecules in a continuous vector space (encoder); for generation, an inverse step (decoder) converts a continuous vector in latent space to a valid molecule. This is the classical encoder-decoder setup as in a standard VAE (Kingma and Welling, 2013). In this work, we augment the VAE framework with a mechanism to learn property specific subspaces — i.e. for each molecular property $j$, we learn property aligned transformations $\mathcal{T}_j : \mathbb{R}^d \longrightarrow \mathbb{R}^k$ where $k \ll d$ and $j$ indexes a molecular property.

The intuition behind the methodology is that while a variational autoencoder (VAE) provides a flexible latent embedding of molecular structures, the geometry of this space is not automatically aligned with variation in molecular properties of interest, unless explicitly trained through supervision or guidance. Several existing approaches in the literature address this by training auxiliary prediction tasks directly on top of the generative latent space, effectively endowing it with semantic meaning. A key perspective, widely recognised in literature is that although the latent space may be globally high-dimensional and entangled, meaningful variations often concentrates along oblique, lower-dimensional manifolds with some residual variation dispersed along several entangled dimensions.

This makes property prediction and optimisation opaque, as gradients can be dispersed across nuisance directions only weakly aligned with coherent property variation. Traversing the chemical latent space, however, is a core operation in chemical generative modelling underpinning guided exploration of novel molecules.

In our framework, we propose a technique to disentangle property-specific variation in the global high-dimensional latent vectors into a transformed coordinate system. By non-linearly warping the latent coordinates, we target a representation where property similarity contracts distances, bringing embeddings of molecules close together. To achieve semantic organisation in the transformed coordinate space, the parameters of the warping function are optimised through a novel loss construction which targets this alignment. This loss function not only encourages semantic organization by pulling molecules with similar properties closer together in the transformed space but also encourages a consistent direction of property-specific variation in the latent space. We also provide techniques around property prediction in the warped space and the decoding protocol to mathematically lift embeddings from the warped space to the original global latent space for decoding.

**Representation Syntax** Most of the literature on machine learning based chemical design for the string representation of molecules uses SMILES strings (Weininger, 1988) — a line notation method which encodes molecular structure using short ASCII strings. However, the SMILES representation has two critical limitations. First, it is not designed to capture molecular similarity: molecules with almost identical structure can have markedly different SMILES (Jin et al., 2018). Second, it is not robust on its own, which means that generative models are likely to produce strings that do not represent valid molecules. Hence, the latent space of DGMs trained on SMILES strings can potentially have large dead zones where none of the points sampled in the region decode to valid molecules. To overcome these issues, we train our model on an alternative string representation for molecules introduced in 2020 (Krenn et al., 2020) that guarantees $100\%$ robustness — SELF-referencing embedded string (SELFIES). We do not dive deep into technical construction aspects of the SELFIE syntax in this work. At a high-level one of the difficulties of working with SMILES is the nested bracket closures which appear frequently in the SMILES notation and introduce syntactic fragility. For instance, consider the smiles string `CCCc1cc(NC(=O)CN2C(=O)NC3(CCC(C)CC3)C2=O)n(C)n1`. The SELFIE translation uses a formal Chomsky type-2 grammar or a context-free grammar and gets rid of the non-local characteristics. The molecule above is translated to "[C][C][C][C][C][=C][Branch2][Ring1][=C][N][C][=Branch1][C][=O][C][N][C][=Branch1][C][=O] [N][C][Branch1][O][C][C][C][Branch1][C][C][C][C][Ring1][#Branch1][C][Ring1] [N][=O][N] [Branch1][C][C][N][=Ring2][Ring1][#Branch1]". We tokenize the SELFIE syntax to represent molecules in our generative model. The chemical properties we use in experiments are hydrophobicity (logP), synthetic accessibility (SAS) and quantitative estimate of drug-likeness (QED) score, all of which are represented through continuous values and widely available for large drug libraries. However, our framework does not rely on fully labeled datasets; it naturally accommodates partial supervision, as the alignment losses are computed only with respect to the observed property labels within each batch.

**Related Work** The fundamental idea of using a shared latent space which serves as input to an auxiliary model for property prediction was first proposed in Gómez-Bombarelli et al. (2018) in the cheminformatics space. The motivation for training the generative architecture along with an auxiliary neural net for property prediction jointly can be attributed to encoding property similarity in the latent space. However, this technique treats the latent space as a single monolithic embedding and enforces regularisation with respect to different properties on the entire latent space. As a result, it lacks the ability to disentangle or isolate the dimensions most aligned to specific molecular properties. Since then, several works have relied on various techniques to induce a semantically regularised latent space (Jin et al., 2018; Kang and Cho, 2018; Popova et al., 2019; Zang and Wang, 2020; Tevosyan et al., 2022). Further, several other works additionally focus on adapting conditional VAE architectures (Kang and Cho, 2018; Lim et al., 2018; Ang et al., 2023; Joo et al.) for property conditioned molecular generation. A parallel line of work explores Bayesian optimisation in the latent space of generative models using Gaussian process surrogates to model property variation (Griffiths and Hernández-Lobato, 2020; Eriksson et al., 2019; Maus et al., 2022; Lee et al., 2024; Tripp et al., 2020). While these methods differ in how they address high-dimensionality and acquisition through trust regions, projection techniques, or kernel selection techniques, they typically treat the latent space as an indivisible embedding space. Our work departs from this by disentangling the high-dimensional latent space into a transformed coordinate space post-hoc; the warping function acts as a spotlight

on property specific variation and decoding enabling optimisation in low-dimensions mitigating the curse of dimensionality.

We summarise the key innovations over existing work below:

- **Seq2seq Transformer VAE on SELFIES.** We present state-of-the art results for reconstruction and several other empirical metrics for this generative architecture trained with SELFIES[1].

- **Learning property aligned coordinate space via differentiable warping function.** This enables gradient-based traversal and decoding aligned to property variation in a compressed lower dimensional space.

- **Property optimization through subspace traversal** We present a principled approach for optimising properties in low-dimensional transformed coordinate space mitigating the curse of dimensionality subsequently lifting the optimised point back into the full latent space for decoding in a numerically stable fashion.

**Experimental scope**   We demonstrate through several experiments, both quantitative and qualitative, the general performance of our model on publicly available benchmark datasets for molecular generation. More specifically, we show that we can achieve high-fidelity reconstructions and sensible interpolations in latent space. We qualitatively show semantic regularisation in the transformed space and the directional gradient with respect to properties. Through neighbourhood visualisations (also seen in Jin et al. (2018); Kusner et al. (2017); Zang and Wang (2020)) we show that the structure of the molecules varies smoothly as a function of distance.

## 2   METHODS

Our generative framework has two components: 1) A sequential Transformer VAE which forms the backbone for reconstructing the molecular sequence structure and is jointly trained with a baseline property predictor in the original latent space 2) a non-linear warping function to map to a transformed and compressed coordinate space with a property alignment loss function with an adjoining linear head for improved property prediction.

### 2.1   TRANSFORMER-BASED VARIATIONAL AUTOENCODER

The generative backbone features an encoder-decoder style architecture, we use transformer layers for both the encoder and decoder. The encoder takes as input a fixed length, tokenised molecular string, $\boldsymbol{x}_{1:T} = [\boldsymbol{x}_1, \boldsymbol{x}_2, \ldots, \boldsymbol{x}_T]$ where $T$ is determined by the length of the longest molecule in the training dataset, $\boldsymbol{x}_{1:T} \in \mathbf{X}$. Shorter molecule strings are padded up with a reserved token to bring them up to $T$ dimensions. Each token is embedded and augmented with a positional encoding, then processed by a stack of self-attention blocks to produce contextual states $H_{1:T} \in \mathbb{R}^{T \times D}$ where $D$ is the embedding dimension. We obtain a sequence summary $\boldsymbol{h} = \frac{1}{T} \sum H_t \in \mathbb{R}^D$ by mean pooling which is used to parametrise the global latent $\mathbf{z}$ governed by the variational approximation,

$$q_\phi(\mathbf{z}|\boldsymbol{x}_{1:T}) = \mathcal{N}(\boldsymbol{\mu}_z(\boldsymbol{x}_{1:T}), \mathrm{diag}(\boldsymbol{\sigma}_z^2(\boldsymbol{x}_{1:T}))), \text{where}, \boldsymbol{\mu}_z = W_\mu \boldsymbol{h} + \boldsymbol{b}_\mu, \log \boldsymbol{\sigma}_z^2 = W_{\log v} \boldsymbol{h} + \boldsymbol{b}_{\log v} \tag{1}$$

As in a standard VAE we assume a standard Normal prior for $\mathbf{z} \sim \mathcal{N}(0, I_d)$.

The decoder is a Transformer with causal self-attention to broadcasted latent memory; more concretely we project the latent $m = W_z \mathbf{z} + b_z \in \mathbb{R}^D$ and repeat it across time as $M = [m, m, \ldots, m] \in \mathbb{R}^{T \times D}$ which serves as keys/values at every decoder layer. With teacher forcing, the likelihood factorizes as,

$$p(\boldsymbol{x}_{1:T}|\mathbf{z}) = \prod_{t=1}^{T} p(\boldsymbol{x}_t|\boldsymbol{x}_{<t}, \mathbf{z}), \tag{2}$$

$$p(\boldsymbol{x}_t \mid \boldsymbol{x}_{<t}, \mathbf{z}) = \mathrm{Cat}(\boldsymbol{\pi}_t), \quad \boldsymbol{\pi}_t = \mathrm{Softmax}(f_\psi(\boldsymbol{x}_{<t}, \mathbf{z})), \tag{3}$$

---

[1]Transformer VAEs are not new and have been used in other seq2seq tasks, its application and evaluation in this setting has never been presented before to the best of our knowledge.

where $f_\psi$ is the transformer decoder with parameters $\psi$ that takes as input the previously decoded tokens $\boldsymbol{x}_{<t}$ along with a broadcast or concatenated version of $\mathbf{z}$ at each layer. The decoder self-attention uses a causal mask $\mathcal{M}$ (forbids attending to future positions) and a key-padding mask $\mathcal{P}$ (ignores pads). At the head level, attention is $\text{Attn}(Q, K, V) = \text{softmax}\left(\frac{QK^\top}{\sqrt{D}} + \mathcal{M} + \mathcal{P}\right) V$ for self-attention over targets with $\mathcal{M}$ and cross-attention from targets over latent memory $M$. This design gives the decoder global control from $\mathbf{z}$ at every layer and time step, improves long-range dependency modelling (rings/branches), and allows parallel training over sequence positions.

**Inference and ELBO** The posterior over the global latent variable $\mathbf{z}$ is intractable. We use the variational approximation $q_\phi(\mathbf{z}|\boldsymbol{x})$ above to enable tractable inference through the use of a lower bound.

The joint data marginal likelihood is given by the sum of individual marginal likelihoods, $\log p(\mathbf{X}) = \sum_{n=1}^{N} \log p(\boldsymbol{x}_{1:T}^{(n)})$, and $\boldsymbol{x}_{1:T}^{(n)} \in \mathbf{X}$ represents a single molecular string sequence; each individual marginal likelihood can be lower bounded as:

$$\log p(\boldsymbol{x}_{1:T}^{(n)}) \geq \mathcal{L}(\Phi, \psi; \boldsymbol{x}_{1:T}^{(n)}) = \mathbb{E}_{q_\phi(\mathbf{z}|\boldsymbol{x}_{1:T})} \left[ \sum_{t=1}^{T} \log p(\boldsymbol{x}_t^{(n)}|\boldsymbol{x}_{<t}, \mathbf{z}) \right] - \text{KL}(q_\phi(\mathbf{z}|\boldsymbol{x}_{1:T}^{(n)})||p(\mathbf{z})), \tag{4}$$

where $\Phi = (\phi_1, \phi_2)$ are the encoder parameters and $\psi$ are the decoder parameters. The variational and generative parameters $(\Phi, \psi)$ are learnt by maximising the joint evidence lower bound. In practice we optimize a padding-aware cross-entropy for the likelihood term, with a causal mask and $\beta$-annealed KL.

Note: The terminology "global" latent variable refers to time-step independent and the fact that $\mathbf{z}$ represents an information bottleneck for the entire sequence $\boldsymbol{x}_{1:T}$; this is to distinguish it from models which use latent variables per time-step $\mathbf{z}_t$ (Chung et al., 2015). The motivation behind this design is to enable latent space optimisation on a global latent space where each latent variable $\mathbf{z}$ (Gaussian ellipsoid in $d < T$ dimensional space) represents the compression of an entire molecular string sequence (a single molecule).

The generative model is jointly trained with a supervised property predictor network $f_\theta(\mathbf{z}) = \mathbf{y}$ on the global high-dimensional latent space to complete the Baseline model for Stage 1. Below we describe the precise framework for learning Stage 2 of learning the warped spaces using transformations $T_j$.

## 2.2 Warping the Latent Space for Property Alignment

After training the generative backbone, we learn property specific transformations,

$$T_j : \mathcal{Z} \longrightarrow \mathcal{U}, \text{ where } \mathcal{Z} \subset \mathbb{R}^d, \mathcal{U} \subset \mathbb{R}^k, k \ll d \tag{5}$$

that map global latents $\mathbf{z} \in \mathcal{Z}$ to warped coordinates $\mathbf{u}_j = T_j(\mathbf{z})$. Each $T_j$ is implemented as a 2-layer GELU MLP trained with an alignment loss over all pairs of molecules in a given batch as,

$$\mathcal{L}_{\text{align}}^{(j)} = \frac{1}{|\mathcal{P}|} \sum_{(a,b)\in\mathcal{P}} \left( \|\mathbf{u}_j^a - \mathbf{u}_j^b\|_2^2 - \alpha_j \|y_j^a - y_j^b\|_2^2 \right)^2, \quad |\mathcal{P}| = \binom{B}{2} \tag{6}$$

where a batch is defined by tuples $\{(\mathbf{z}^a, y_j^a)\}_{a=1}^{B}$ denoting the global latent embedding and a scalar property $y_j$ per molecule. This loss makes the geometry of the warped coordinate $\mathbf{u}j$ mirror the magnitude of the property difference $y_j$. The scale $\alpha_j$ matches units so the model is free to warp: it can contract regions where $y_j$ varies little and expand where $y_j$ changes rapidly, while preserving a monotone relation between property difference and embedding distance. Averaging the squared residuals over the batch pairs produces a smooth objective that aligns pairwise distances in $\mathbf{u}_j$- space with pairwise dissimilarities in property space, thereby turning $\mathbf{u}_j$ into a coordinate vector along which similar or aligned embeddings implies closeness in property values $y_j$.

Note that training only the above distance alignment term with a free-scale $\alpha$ has a degenerate solution where the warping collapses all the $\mathbf{u}_j$'s to a constant, making the distances $||\mathbf{u}_j^a - \mathbf{u}_j^b|| \longrightarrow 0$ and if $\alpha_j$ is fit/estimated from the batch then the optimum $\alpha_j^* = 0$, hence, the global minimum collpases the warped embedding $\mathbf{u}_j$. In order to alleviate this degeneracy, we fix $\alpha = 1$ and add a *spread* constraint on $\mathbf{u}$.

**Covariance whitening**  In order to keep $\mathbf{u}_j$ non-collapsed, smooth and well-conditioned we penalise collapse by adding a covariance whitening loss given as,

$$\mathcal{L}_{\text{cov}}^{(j)} = \big\| \widehat{\text{Cov}(u_j)} \, - \, I_{k_j} \big\|_F^2 \tag{7}$$

where we penalise how far the unbiased batch covariance of the warped coordinates are from identity. Overall, it makes the learned $\mathbf{u}_j$ space have unit variance along each axis and zero cross-correlations. This term also help the warped coordinates match the VAE prior and helps with lifting them back to the global latent vectors by keeping the optimisation stay on-manifold.

**Linear head**  For each property $j$ we also fit a linear head on the warped coordinate $\mathbf{u}_j$ as, $\hat{y}_j(\mathbf{u}_j) = w_j^\top \mathbf{u}_j + b_j$. The warping function $T_j$ is trained to concentrate the property-relevant variation along a small number of coordinates. A linear map is sufficient and preferable as it yields a single monotone direction $w_j$ to traverse for optimising a property. Critically, a linear map gives a principled closed-form ascent direction for optimization (see section 2.3) in the warped coordinate space. During training we just use a simple MSE loss for the linear head per property,

$$\mathcal{L}_{\text{mse}}^{(j)} = \frac{1}{B} \sum_{a=1}^{B} \big( \hat{y}_j(\mathbf{u}_j^a) - y_j^a \big)^2 \tag{8}$$

Hence, the overall loss for stage 2 for a given property is,

$$\mathcal{L}_{\text{warp}}^{(j)} = \mathcal{L}_{\text{align}}^{(j)} + \lambda_{\text{cov}} \mathcal{L}_{\text{cov}}^{(j)} + \lambda_{\text{mse}} \mathcal{L}_{\text{mse}}^{(j)} \tag{9}$$

where $\lambda_{\text{cov}}$ and $\lambda_{\text{mse}}$ are hyperparameters which are set to pre-defined schedules as training progress in order to control the influence of each of the terms on the overall loss.

## 2.3 PROPERTY OPTIMISATION AND DECODING PROTOCOL

Given a warped property-aligned coordinate space $\mathbf{u}_j$ and a trained linear head $\hat{y}_j(\mathbf{u}_j) = w_j^\top \mathbf{u}_j + b_j$, we can score candidates via the objective,

$$U(\mathbf{u}_j) = w_j^\top \mathbf{u}_j \, - \, \gamma \|\mathbf{u}_j\|_2^2, \qquad \gamma > 0 \tag{10}$$

The first term promotes movement along the monotone property axis $w$; the second is a quadratic trust-region that discourages departures to large $\|\mathbf{u}\|$ by exerting a pull towards the origin (it is equivalent to having a standard Gaussian prior on $\mathbf{u}$). This objective has a closed-form maximizer,

$$\nabla_u U(\mathbf{u}_j) = w - 2\gamma \mathbf{u}_j = 0 \implies \mathbf{u}_j^\star = \frac{1}{2\gamma} w \tag{11}$$

i.e., the optimum lies on-axis and its magnitude is set by $\gamma$. Without the penalty ($\gamma = 0$), gradient ascent would shoot off to infinity along $+w$, leaving the support of the learned geometry and yielding unreliable decodes to the global latent space. The gradient ascent updates in $\mathbf{u}$-space is given by,

$$\mathbf{u}_j^{t+1} = \mathbf{u}_j^t \, + \, \eta \big( w \, - \, 2\gamma \mathbf{u}_j^t \big) \tag{12}$$

where $\eta$ is the learning rate.

**Decoding Protocol**  Ultimately, we need to lift molecules back into the global latent space in order to decode them with the transformer decoder. Importantly, $T_j$ is not invertible, it is generally non-injective (many-to-one) because $k \ll d$ and the warping transformation deliberately contracts/identifies nuisance directions that are irrelevant to $y_j$ This non-injectivity is not undesirable as itt denoises and concentrates property-relevant variation onto a small set of coordinates. For decoding, since $T_j^{-1}$ does not exist, we lift from $\mathbf{u}_j^*$ back to the backbone latent by solving a small optimization,

$$\mathbf{z}^\star = \arg\min_{\mathbf{z} \in \mathcal{Z}} \big\| T_j(\mathbf{z}) - \mathbf{u}_j^\star \big\|_2^2 \, + \, \lambda \big\| \mathbf{z} - \mathbf{z}_0 \big\|_2^2. \tag{13}$$

where the second term biases the solution towards a soft radius around the starting point $\mathbf{z}_0$. $\lambda$ can be set to zero but it is particularly useful ($\lambda > 0$) if the goal is to preserve molecular structure while nudging a property. In order to mitigate local optima we perform multi-restart latent optimization in practice from a diversified set of starting points in the global latent space. However, in order to bias optimisation towards favourable regions we can select starting points from molecules known to have high property values $y_j$.

## 3 EXPERIMENTS

**Dataset and pre-processing**   We use two widely used datasets for our experimental evaluation – the ZINC 250k dataset comprises 250K randomly sampled molecules from the ZINC database (Sterling and Irwin, 2015) containing over 120 million "drug-like" compounds. The QM9 dataset (Ruddigkeit et al., 2012) contains about 134K small molecules with fewer than nine heave atoms (C, N, O, F). Both datasets are provided in the canonical SMILES representation. Hence, we translate the string representation to the SELFIES syntax using their publicly available python package called selfies. The package provides several convenience methods to convert SELFIES strings to one-hot-encodings or integer encodings. For QM9, ZINC and PubChem we work with integer encoding. Since the molecules come in different sequence lengths, we pad each of the encodings to a max length (length of the longest sequence) using a padding token `[nop]` which is a part of the alphabet.

### 3.1 RECONSTRUCTION, VALIDITY AND SAMPLING

We report reconstruction accuracy alongside several standard metrics commonly used to evaluate the performance of chemical generative models on the ZINC 250k dataset (see table 1). We employ an 80/10/10 split for training, validation, and testing. Our generative model demonstrates performance comparable to other state-of-the-art VAE-based approaches for molecular string generation. Notably, using the SELFIES representation allows us to circumvent the syntactic fragility of SMILES and ensures 100% validity in generated molecules. Furthermore, our Transformer-based VAE achieves 100% novelty—measured as the fraction of unconditionally generated molecules not present in the training data, and high uniqueness, which accounts for duplicates. The statistics for our method are computed on 100,000 samples drawn from the $\mathcal{N}(0, 1)$ prior post-training. We also report Monte Carlo (MC) reconstruction accuracy, in which a fixed batch of molecules is encoded and decoded $n$ times, and the reconstruction accuracy is computed as the fraction of exact matches between the generated sequences and the original inputs.

| Metrics | Test Reconstruction Accuracy | | |
|---|---|---|---|
| Datasets | QM9 | ZINC250K | Pubchem |
| Dataset size | 132K | 250K | 500K |
| Accuracy (%) | 99.3% | 99.67% | 99.8% |

It is worth noting that the average reconstruction accuracy of our method, calculated as the fraction of sequence tokens that match the ground truth, averaged across all sequences in a held-out set is on par with the top accuracies reported in literature.

Table 1: Summary of generative performance across a range of metrics for the ZINC 250k dataset for a subset of popular methods reported in literature. Some of the statistics reported are compiled directly from (Kusner et al., 2017; Jin et al., 2018) for the string representation and (Zang and Wang, 2020) for the graph representations.

| Models | Representation | % Validity | % Novelty | %Uniqueness | % MC Reconstruction |
|---|---|---|---|---|---|
| Chem-VAE (Gómez-Bombarelli et al., 2018) | SMILES | 0.7 | n/a | n/a | 44.6 |
| Grammar-VAE (Kusner et al., 2017) | SMILES | 7.2 | n/a | n/a | 53.7 |
| SD-VAE (Dai et al., 2018) | SMILES | 43.5 | n/a | n/a | 76.2 |
| JT-VAE (Jin et al., 2018) | Graphs | 99.8 | 100 | 100 | 76.7 |
| MoleculeRNN (Popova et al., 2019) | Graphs | 100 | 100 | 99.89 | n/a |
| MoFlow (Zang and Wang, 2020) | Graphs | 100 | 100 | 99.9 | 100 |
| GraphNVP (Madhawa et al., 2019) | Graphs | 42.6 | 100 | 94.8 | 100 |
| GRF (Honda et al.) | Graphs | 73.4 | 100 | 53.7 | 100 |
| Transformer VAE (ours) | SELFIES | 100 | 100 | 99.9 | 87.1 |

For some additional context, note that some of the recent graph based approaches like Zang and Wang (2020); Honda et al.; Madhawa et al. (2019) boast 100% MC reconstruction rate, however their empirical running times for this dataset for a 200 epoch training schedule are reported to be 22 hours for MoFlow (Zang and Wang, 2020), 38.4 hours for GraphNVP(Madhawa et al., 2019) and over 120

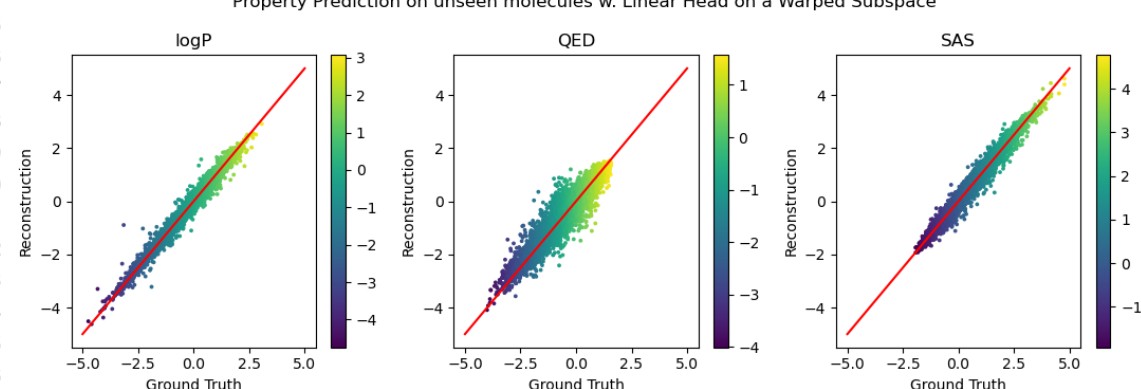

Figure 1: Ground-truth vs. predicted molecular properties on a held-out ZINC250K test set using a linear head trained on a warped latent space. Each panel corresponds to one property (logP, QED, SAS). Points show model predictions ("Reconstruction") against ground truth; the red line denotes the identity ($y=x$). Points are coloured by the ground-truth value to visualise any range-dependent effects. Across properties, predictions align closely with $y=x$ with no evident systematic bias across the value range, indicating that a simple linear head on the warped subspace generalises to unseen molecules.

hours for GRF (Honda et al.) on a GeForce RTX 2080 Ti 16 core GPU machine. Our model trains in under 2 hours on a NVIDIA RTX A6000 utilising a single core. We note that this is not a like-for-like comparison in terms of compute configuration. Differences in hardware, implementation details, and model architecture all influence runtime. Nonetheless, these figures highlight the practical efficiency of some of these approaches.

Details about the generative architecture and learning parameters for our transformer based VAE can be found in the appendices.

## 3.2 PROPERTY PREDICTION

In this section we report the mean squared error of the property prediction task on a held-out test dataset from ZINC250 where we have the ground truth properties. In fig. 1 we show the values of the ground truth property v reconstruction for each of the properties predicted by the linear head on a 4D warped space ($k = 4$). By shading the points by their ground truth values we can observe that there is no correlation between the bias of the estimate and the ground truth value. The reconstructions shown are just computed as $\hat{y}_j = w^T \mathbf{u}_j + b_j$ for each property $j$. Despite its low capacity, the linear head achieves surprisingly competitive accuracy compared to the canonical non-linear baseline predictor trained on the entangled high-dimensional latents (see table 2) while yielding a highly compressed interpretable space. Across properties we observe high rank correlation (Spearman $\rho$) and calibration slopes near 1, confirming that the warp concentrates property relevant variation so that a linear readout suffices. The identical plot for the baseline predictor is in the appendices. All property values were standardised to zero mean and unit variance before training and we report the RMSE on standardized values.

Table 2: Test RMSE scores for property prediction using the warped coordinate space v. the baseline model.

| Metric | | RMSE: Test (Train) | | |
|---|---|---|---|---|
| Model ($\downarrow$) | Head | logP | QED | SAS |
| Baseline | Non-linear | 0.1381 (0.1203) | **0.2546**(0.1765) | 0.1556 (0.1383) |
| Warped | Linear | **0.1364** (0.118) | 0.2560 (0.144) | **0.1474** (0.1311) |

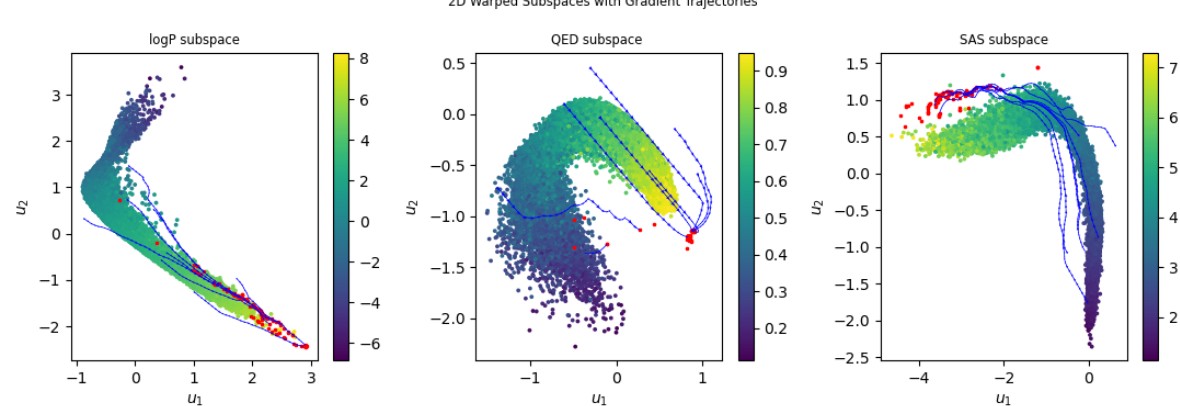

Figure 2: Two-dimensional slices of the learned 4D warped subspaces for logP, QED, and SAS, with gradient-based optimisation trajectories (blue) and converged optima (red). Points are coloured by ground-truth property values. The optimiser converges to high-scoring regions of the manifold; a projection-to-radius step $\mathbf{u} \leftarrow \mathbf{u}_{\mathrm{nn}} + \min\left(1, r/\|\mathbf{u} - \mathbf{u}_{\mathrm{nn}}\|\right)\left(\mathbf{u} - \mathbf{u}_{\mathrm{nn}}\right)$ enforces a trade-off between exploration (larger $r$) and fidelity to the data manifold (smaller $r$). $\mathbf{u}_{\mathrm{nn}}$ denotes the nearest neighbour of the current optimisation iterate among the set of training embeddings.

## 3.3 WARPED SUBSPACES

In fig. 2 we visualise a 2D slice of the 4D warped space learnt for each of the molecular properties. Each panel shows the embedding of molecules in the corresponding warped space, with points coloured by the ground-truth property value. The geometry of each subspace reflects the non-linear structure that the warping captures, revealing smooth property gradients. To evaluate whether these subspaces can be used for targeted molecular optimisation, we further perform gradient-based ascent within each subspace using the linear property heads. Starting from random initial molecules, we follow gradient trajectories (blue curves) that traverse the warped coordinates toward regions of higher property value. The final optimised points are marked in red, corresponding to the top-$\mathbf{u}$ configurations identified by the optimiser. Because the head is optimised on observed $\mathbf{u}$'s while the ascent updates are free-form, trajectories can drift slightly off the empirical manifold. To mitigate this, after each gradient step we apply a projection-to-radius (a trust-region update) that snaps $\mathbf{u}$ back toward the nearest sampled point. Empirically, decreasing the radius tightens the trust region keeping trajectories close to the data manifold (with a smaller radius the optimiser would have converged to points on the data manifold, the choice of the radius determines a trade-off between exploration and playing safe). Importantly, the converged points consistently land in high-scoring regions of the manifold, where neighbouring molecules already exhibit elevated property values. This demonstrates that the warped subspaces not only disentangle variation in individual properties but also provide smooth and navigable landscapes.

## 3.4 QED PROPERTY OPTIMISATION

In fig. 3 we show 5 molecules obtained by decoding the top optimised points from the warped subspace with respect to QED. The generated structures are all novel relative to the training data and achieve QED scores above 0.94 placing them among the highest reported values in literature. Interestingly, despite the diversity of the scaffolds, the associated SAS and logP values remain relatively stable across these top candidates, suggesting that the optimisation in the warped QED subspace can selectively improve drug-likeness without substantially perturbing synthetic accessibility (SAS) or lipophilicity (logP).

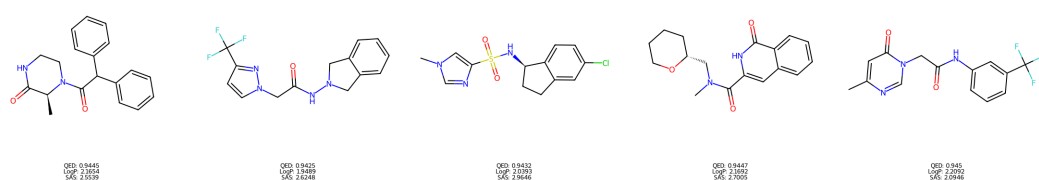

Figure 3: Top molecules decoded from optimised points in the warped QED subspace. All candidates are novel and achieve QED > 0.94. Notably, SAS and logP remain nearly unchanged (see appendices) across diverse scaffolds, evidence of *directional consistency*: ascent along the QED-aligned direction in the warped space leaves orthogonal property components approximately invariant, reflecting coherent control.

It is interesting to note that property optimisation in the baseline model for QED yielded a top score of 0.9208 with 100 restarts. The top QED score in the training data is 0.948. As a comparison JT-VAE (Jin et al., 2018) reported a top score of 0.925 in the QED optimisation task while the graph methods (Popova et al., 2019; Zang and Wang, 2020) recover molecules which achieve the top score of 0.948.

## 4 DISCUSSION, LIMITATIONS, SCOPE

The aim of this work is to contribute an effective technique to search and traverse chemical latent spaces for the small-molecule drug modality. We conduct our experiments with the recently developed (Krenn et al., 2020) SELFIE syntax for molecular string representation to side-step an important failure mode of the commonly used SMILES representation. Beyond, a performant generative backbone we showed that the auxiliary warping step is extremely effective for learning a compressed transformed coordinate space with consistent direction of property variation and a linear property prediction head which surpasses the baseline method.

While we only use SELFIES for the molecular representation, the methodology and proposed alignment techniques can be integrated with generative models built for other string representations of molecules like extended connectivity fingerprints (ECFP) (Rogers and Hahn, 2010), SMILES or DeepSMILES (O'Boyle and Dalke, 2018); or graph representations (Zang and Wang, 2020; Popova et al., 2019) as well as just any generative modelling setting where there is a core input along with a measured context or condition to align with. It is important to note that the computational overhead of training the warping function through alignment loss is quadratic in batch size (similar to contrastive loss and other pairwise geometry objectives) with overall complexity $\mathcal{O}(B^2 K)$ where $B$ is the batch size and $K = \dim(u)$. A limitation of the overall design is that our whole approach encourages the formation of property-specific semantic subspaces, where 'property' denotes either a scalar observable or a composite functional (e.g., the penalized logP, pLogP), formed by combining logP with structural penalty terms. We do not yet explicitly model cross-property correlations potentially overlooking shared latent structure.

However, one advantage is that there are no practical limits to the number of property specific warped spaces that can be simultaneously learnt in this manner. This contrasts with more commonly used approaches, where regularising across multiple properties creates competition for limited latent dimensions and may lead to interference effects leading to diminished regularisation or even worse, compromised reconstruction. More work is needed to understand their behaviour.

This work can be extended in several ways. An important direction for future work is extending this framework to settings with partial feedback where the property space is only sparsely observed. In real-world molecular datasets, it is often the case that only a subset of properties are available for each compound. Application to active learning settings such as Bayesian optimisation is also an important experimental direction. While we only experiment with a specific choice of warping functions and property predictors there could be other choices that may work well such as Gaussian processes where one can inherently control smoothness attributes of the input space through the choice of the kernel function. Finally, a true test of these models is to assess their generative capabilities on genuine target based lead discovery, for instance, designing a lead molecule that binds to a target with high affinity and is simultaneously retrosynthetically accessible. Our framework with the ability to move along an coherent direction of variation can be strong advancement on existing techniques.

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

# 5 SUPPLEMENTARY

## 5.1 BATCH SIZE AND PAIRWISE METRICS

The alignment loss is computed over all unordered pairs of a mini-batch, giving $|\mathcal{P}| \approx B^2$ pairs for batch size $B$. In the idealized case of independent pairs, the variance of the stochastic gradient scales as,

$$\mathrm{Var}[\nabla \mathcal{L}_{\mathrm{align}}] \;\propto\; \frac{\sigma^2}{|\mathcal{P}|} \;\approx\; \frac{\sigma^2}{B^2}\,.$$

Thus, doubling the batch size would in principle quarter the gradient variance. In practice the pairwise terms are correlated, so the empirical scaling falls between $1/B$ and $1/B^2$. Nonetheless, increasing batch size consistently improves stability and often reduces the number of steps needed to reach a given training target.

# 6 ADDITIONAL RESULTS

## 6.1 BASELINE PROPERTY PREDICTION

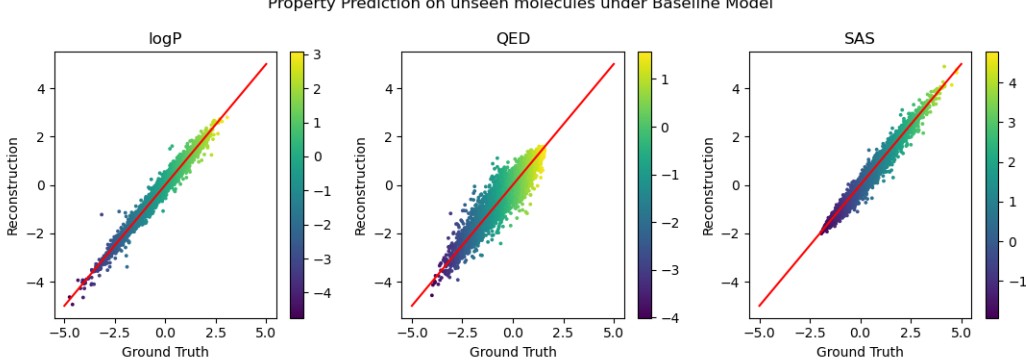

Figure 4: Property prediction with the baseline model on the global latent space.

## 6.2 QED TOP MOLECULES

The SMILES string of the top QED molecules shown in the main section are given below along with the corresponding SAS and logP scores which were computed in RDkit (as these molecules were not in the training data).

C[C@H]1C(=O)NCCN1C(=O)C(C2=CC=CC=C2)C3=CC=CC=C3
QED: 0.9444 LogP: 2.1654 SAS: 2.5538

O=C(CN1C=CC(C(F)(F)F)=N1)NN2CC3=CC=CC=C3C2
QED: 0.94254 LogP: 1.9489 SAS: 2.6247

CN1C=NC(S(=O)(=O)N[C@@H]2CCC3=CC(Cl)=CC=C32)=C1
QED: 0.94315 LogP: 2.0392 SAS: 2.9646

CN(C[C@H]1CCCCO1)C(=O)C2=CC3=CC=CC=C3C(=O)[NH1]2
QED: 0.94465 LogP: 2.1692 SAS: 2.7004

CC1=CC(=O)N(CC(=O)NC2=CC=CC(C(F)(F)F)=C2)C=N1
QED: 0.9450 LogP: 2.2092 SAS: 2.0946

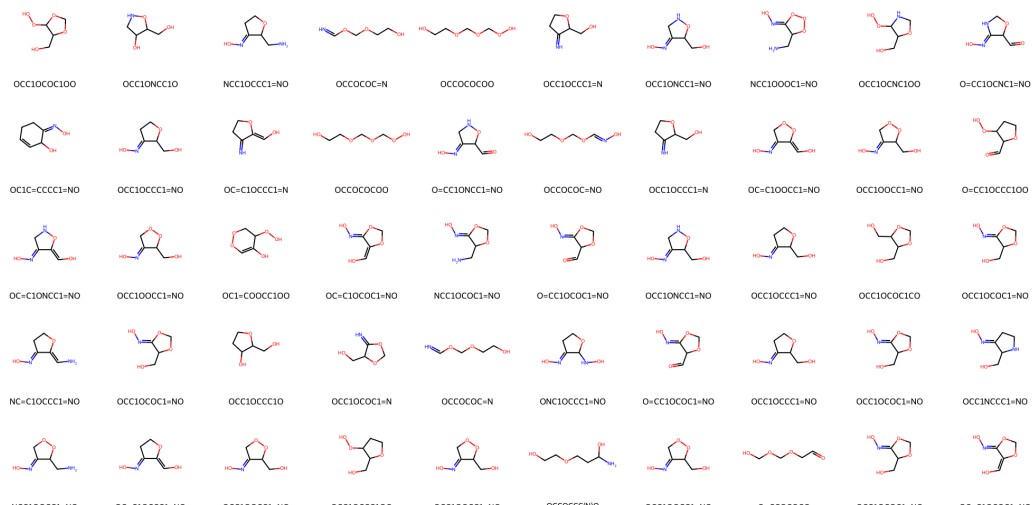

Figure 5: Samples from an $\epsilon$ neighbourhood of a seed molecule (top-left) from the QM9 dataset (fewer than 9 heavy atoms).

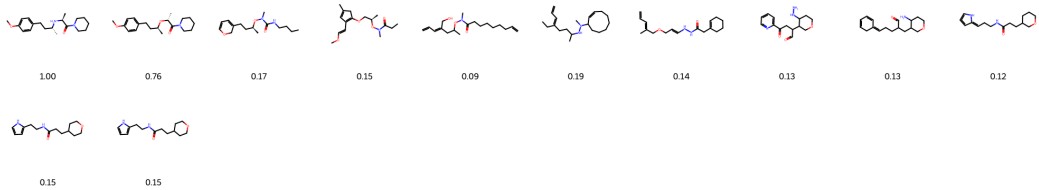

Figure 6: Interpolation in the global latent space, 10 Euclidean steps, the scores denote the Tanimoto similarity

## 6.3   NEIGHBOURHOOD VISUALISATION AND INTERPOLATION

In order to examine the decoder smoothness of our architecture, we show interpolations and traversals with pubchem and QM9 data in the global latent space. For interpolations we report the chemical similarity to the seed molecule measured with the Tanimoto kernel (Gower, 1971). One can observe gradual changes in the molecular graph as the similarity co-efficient recedes gradually as a function of distance to the starting molecule.

## 7   EXPERIMENTAL CONFIGURATIONS

**Architecture**   The transformer encoder and decoder were trained with 4 layers and 6 attention heads, with am embedding size of 132 and hidden dimension of 300. The dimension $d$ of the global latent space was 256. The dimension of the warped space was kept at $k = 4$ for all the properties.

The MLP for the baseline property predictor was a 2-Layer GELU network which mapped to a joint property vector of dimension 3.

**Training and schedules**   The generative backbone was trained for 120 epochs with $\beta$ annealing for the KL term given by the schedule: beta_kl = min(0.2, (epoch / 40.0) * 0.2).

For the baseline property predictor, the scaling term is given by:

Figure 7: Interpolation in the global latent space, 40 Euclidean steps, the scores denote the Tanimoto similarity

$$
\lambda(t) = \begin{cases} \lambda_{\min}, & t < 5, \\[2mm] \lambda_{\min} + \dfrac{1 - \lambda_{\min}}{7}\,(t - 5), & 5 \le t < 12, \\[2mm] 1, & t \ge 12, \end{cases}
$$

where $\lambda_{\min} = 1e - 3$ and $t$ is the epoch.

For the $\lambda_{\mathrm{mse}}$, the schedule is given by,

$$
\lambda_{\mathrm{mse}}(t) = \begin{cases} \lambda_{\min}, & t < 3, \\[2mm] \lambda_{\min} + \dfrac{1 - \lambda_{\min}}{12}\,(t - 3), & 3 \le t < 15, \\[2mm] 1, & t \ge 15, \end{cases}
$$

with $\lambda_{\min} = 0.1$

For $\lambda_{\mathrm{cov}}$, the weight for the covariance whitening loss followed the schedule,

$$
\lambda_{\mathrm{cov}}(t) = \begin{cases} \lambda_0, & t < \mathrm{hold}, \\[2mm] \lambda_{\mathrm{floor}}, & t \ge \mathrm{end}, \\[2mm] \lambda_{\mathrm{floor}} + \frac{1}{2}\,(\lambda_0 - \lambda_{\mathrm{floor}})\Big(1 + \cos\big(\pi \frac{t-\mathrm{hold}}{\mathrm{end}-\mathrm{hold}}\big)\Big), & \mathrm{hold} \le t < \mathrm{end}, \end{cases}
$$

with $\lambda_0 = 0.1$, $\lambda_{\mathrm{floor}} = 0.05$, hold=10 and end=25.

The warped transformations were trained for 90 epochs for each property and the maximisations were conducted with 100 restarts for 2000 steps with a step size of 1e-4. The decoding protocol for a single molecule used multiple diverse seeds sampled from the prior or alternatively, preset to specific molecules for testing.

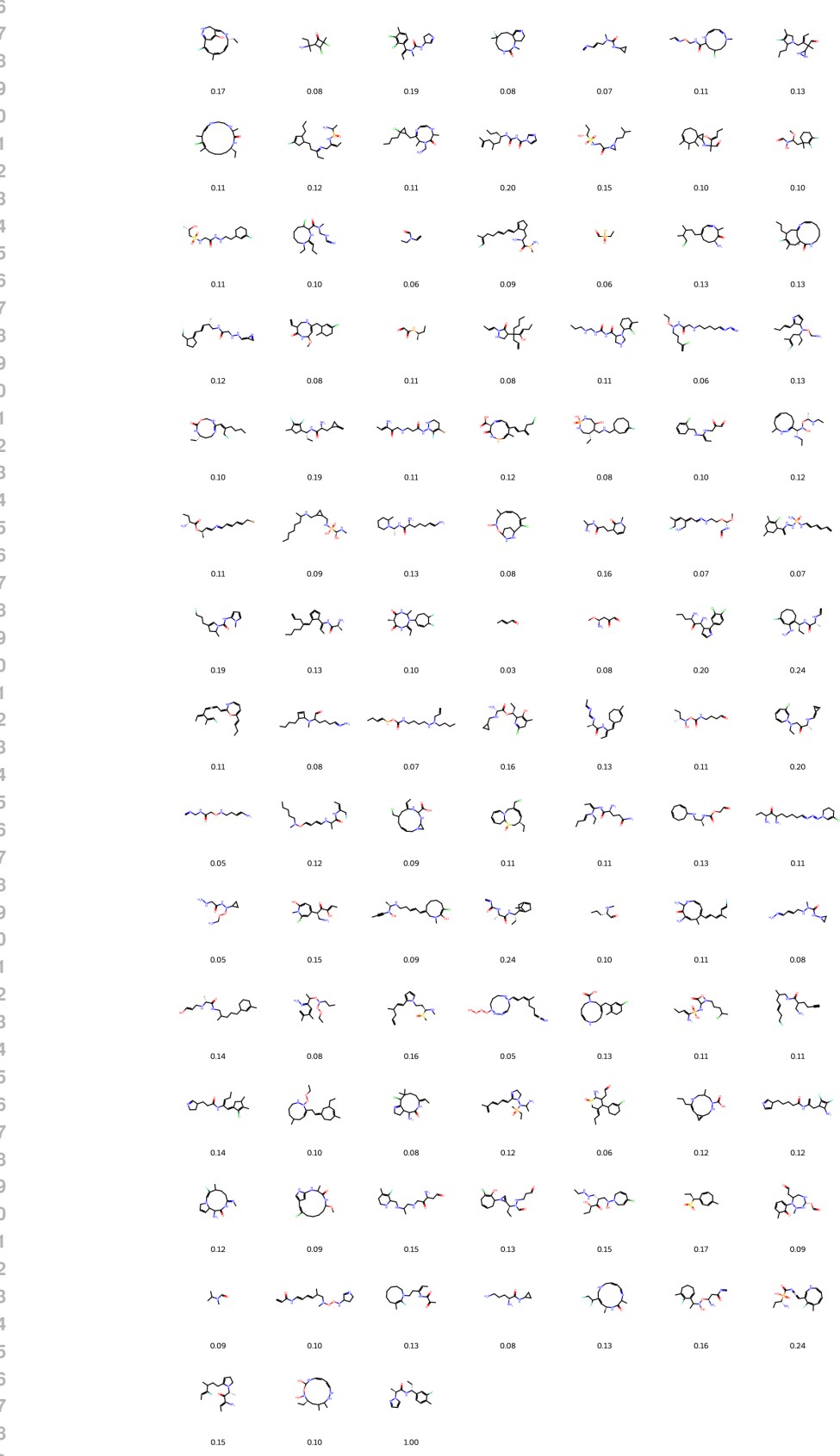

Figure 8: Samples from an $\epsilon$ neighbourhood of a molecule from pubchem data.

