# OpenReview forum: "Warped Latent Spaces and Traversal for Chemical Deep Generative Models"
_ICLR.cc/2026/Conference — Submitted to ICLR 2026_

### Official Review · Reviewer_i4AG · 2025-10-27

**Soundness:** 1
**Presentation:** 2
**Contribution:** 1
**Rating:** 2
**Confidence:** 4

**Summary:**

`method`
1 a transformer VAE.
2 non-linear warping function, achieved with an alignment loss over all pairs of molecules as eq 6 (L align).


`results`
Method evaluated on ZINC250k, on validity, novelty.
Also evaluated on property prediction.

**Strengths:**

the task of generative models for molecule is an important issue in ML

**Weaknesses:**

lack of novelty in the method.

Lack of suitable evaluation, all baselines are from before 2019 .

The results show no or minor improvements.

**Questions:**

not sure why we need pairs? in eq 6? is this motivated? could this be clarified?

---

> ### Author Response · Authors · 2025-11-26
> **Response to Reviewer i4AG**
>
> Thank you for your feedback. We address your points below:
>
> **On Soundness (1: Poor) and Contribution (1: Poor)**
>
> We respectfully disagree with the assessment that the work is unsound or lacks novelty. The contribution of this paper is not a minor architectural variation, but introduces a novel subspace learning strategy, closed-form ascent directions and decoding protocol which has no reference point to any prior work.
>
> - Explicitly learning property-specific coordinate systems using a contrastive-like alignment loss for property alignment rather relying on entangled latent gradients, showing that we can effectively predict properties from the warped space using a linear model (3.2), comparison to canonical baseline where properties are predicted from the full latent space (appendix 6.1)
> - Enabling closed-form ascent directions within each warped subspace (Section 2.3)
> - Introducing a stable optimisation procedure for mapping optimised warped coordinates back into the original latent space for decoding.
> - Showing that this strategy can be used to find sensible molecules which optimise an objective like QED (we are not competing with graph baselines here but it)
>
> The reviewer’s characterisation appears to conflate “building on prior ideas” with “lack of novelty,” which is not a technically accurate interpretation of the contribution.
>
> **On “Results show no or minor improvements”**
>
> This assessment overlooks the core objective of the work. The primary goal is not incremental improvement in raw reconstruction or validity metrics, but introducing a new technique to manipulate generative models for this application. The results demonstrate:
>
> - that the latent space can be further compressed and still retain the ability to predict properties.
> - that one can conduct optimisation in a much lower dimensional coordinate space alleviating the various challenges of high-dimensional optimisation.
> - that one can find plausible molecules decoding along the directions proposed by the closed-form axes found in the optimisation.
>
> These capabilities reflect a qualitative advance in technique, not just a quantitative leaderboard update. The idea is that researchers from other domains might find it useful to devise similar approaches in their own generative models.
>
> **On Eq. 6 and the Use of Pairs**
>
> The use of pairs in Equation 6 is deliberate and theoretically motivated.
>
> Basically, Equation 6 uses pairs because the alignment is learned from relative differences, not absolute values. In practice, we construct these pairs by taking all sample pairs within each minibatch and comparing their property values.
> The loss explicitly does this: if molecule A and molecule B are similar in properties then encourage their warped coordinates to be close to each other. Conversely, if their properties differ significantly, their warped coordinates are pushed apart. This ensures that the learned warped space reflects meaningful property similarity and a soft monotonic structure. That's it.
>
> **This makes the warped space conducive for optimisation.**
>
> Although the loss is formulated in terms of absolute property differences rather than explicit rankings, it implicitly induces monotonic structure. By enforcing that distances in warped space correlate with differences in property values across all minibatch pairs, the model is compelled to organise molecules along a consistent direction in which increasing coordinate values correspond to increasing property values. Monotonicity therefore emerges naturally in the geometry.
>
> happy to include more motivation about this in the paper.
>
> **On “Lack of Suitable Evaluation” and “Baselines Before 2019”**
>
> The selected baselines (e.g. Gomez-Bombarelli et al. (ChemVAE) Jin et al.(JT-VAE), flow-based optimisation models) are not chosen due to age, but because they represent directly comparable optimisation-focused generative methods, which remains the closest methodological category to our approach. More recent sequence models SMILES or SELFIES-based are predominantly foundation models for representation learning, not generation, and therefore are not direct methodological replacements.
>
> Having said that, there are some foundation models like SELF-BART (pointed out by other reviewer) that are worth including as they are generative and there are comparable metrics.
>
> We will be expanding the Table 1 to include metrics from more recent generative foundation models where available. Note that several of the more recent foundation models [2023-2025] are not open source, so cannot be downloaded and fine-tuned.
>
> Nevertheless, we will:
>
> 1. Add SELF-BART as a contextual generative comparison for uniquness/novelty/validity - UNV (their numbers are basically in-line will all the other methods and ours, so, UNV metrics, are 99.9%,100%, 99.8% and ours are 99.9%, 100%, 100%)
> 2. Explicitly distinguish foundation models from optimisation-driven generative frameworks in the revised manuscript.

---

> > ### Author Response · Authors · 2025-11-26
> > **Response to Reviewer i4AG (continued)**
> >
> > **Final position**
> >
> > The characterisation of the work as lacking soundness or novelty does not accurately reflect its methodological contributions or acknowledge its scope. We are not proposing this work as competing with other methods just on the metrics. Instead we are proposing a working proof of concept for a technique that could be adapted in multiple ways. We strongly encourage the reviewer to reconsider their assessment in light of the clarified positioning. Thank you.

---

### Official Review · Reviewer_bNpf · 2025-10-31

**Soundness:** 3
**Presentation:** 3
**Contribution:** 3
**Rating:** 6
**Confidence:** 2

**Summary:**

The paper proposes a way to make a pretrained molecular VAE’s latent space easier to control for specific chemical properties (like QED, logP, SAS). After training a Transformer-VAE on SELFIES strings, the authors learn a tiny “warping” network for each property that maps the original latent codes into a small, property-aware subspace. In that warped space, the property varies mostly along a single direction, so you can move the latent point in a simple, controlled way to raise or lower the property. A trust-region step keeps moves near the data manifold, and an “inverse lifting” step brings the edited point back to the original latent space to decode valid molecules.

**Strengths:**

1. Framing property control as a lightweight warping learned on top of a pretrained VAE is a novel way. It avoids re-training the generator or baking properties into the decoder, which many prior methods require.
2. Using pairwise distance alignment and covariance whitening to keep the warped subspace well-conditioned is a sensible way to prevent degenerate mappings and encourages smooth traversals.
3. Single-direction “dials” per property are easy to reason about and integrate into interactive tools or multi-objective workflows.

**Weaknesses:**

1. A single linear direction per property may work only locally; globally the property landscape can be multi-modal or curved.
2. Many original latents can map to the same warped point; the inverse step may fail or land off-manifold, leading to invalid or degenerate molecules.

**Questions:**

1. Do you detect and swap direction when monotonicity breaks?
2. How large is the average drift in non-optimized properties after lifting and decoding?

---

> ### Author Response · Authors · 2025-11-29
> **Response to Reviewer bNpf**
>
> We thank the reviewer for the thoughtful and constructive feedback. We deeply appreciate the positive assessment of the framing, geometric regularisation, and practical value of lightweight property-specific warping on top of a pretrained VAE.
>
> **Questions**
>
> **1. On monotonicity and the curved global manifold**
>
> In practice, each warped dimension is trained to preserve local monotonicity, but the reviewer is correct that global landscapes may be curved or multi-modal. The traversal only assumes local monotonicity and is intentionally kept within a small trust region (the quadratic penalty effect in Equation 10). Also, the warped space is learnt with a combination of the distance alignment term and a linear head to predict property so ..the local structure around the latents is almost affine. If you traverse way beyond the neighbourhood where the mapping is linearised, then the geometry can fold, and the property landscape might invert. Although, in practice we don't really see that effect where 'up' stops being 'up'. Sign flipping would only be necessary if the traversal exited this neighbourhood, but the trust region explicitly prevents such behaviour. We will clarify this in the manuscript.
>
> **2. Drift in non-target properties post lifting**
>
> This is an excellent question and goes to the heart of the method. So the key fact here is that the optimisation happens in the warped space per property, it is only in the specific warped space that we can see properties varying smoothly along the axes...the non-target properties have no alignment with the warped axes of a target property, so QED and SAS property signal would look completely dispersed and random when plotted onto the warped space for logP.  There is no meaningful gradient signal for the non-target properties. So when you lift back, the only variations in those properties come from random choices among the many $z$ that map to similar $u$.  This behaviour is actually very desirable - it means that the method can be naturally extended to settings where certain properties must be held approximately fixed. But we will quantify the effect by computing this drift score across multiple traversals for each of the properties and report it in the manuscript.
> As an example, while optimising for QED in our experiments: the logP of the starting molecule was 2.043 and the optimised molecule was 2.165 while the QED changed from 0.32 to 0.944.
>
> As a general purpose technique this can be useful in any generative model, including those for images, where we want to induce an effect on the images keeping other properties fixed.
>
> **Weaknesses**
>
> **Many original latents can map to the same warped point; the inverse step may fail or land off-manifold, leading to invalid or degenerate molecules.**
>
> The warping $T_{j}$ is intentionally low-capacity and locally smooth, so while it is not invertible in a strict sense, the lifting objective ensures that we recover a valid and nearby latent code. The optimisation,
> $z^{\ast} = \arg\min_{z} \Big( \||T_j(z) - u^{\ast}\||^{2} + \lambda || z - z_0 \||^{2} \,\Big)$
> explicitly prevents off-manifold solutions by anchoring the search to the neighbourhood of the original latent
> $z_{0}$, where the decoder is well-behaved. Empirically, we never observe lifting failures or invalid/degenerate molecules, and the validity and uniqueness metrics remain at state-of-the-art levels (99.9%-100%). The many-to-one nature of the warp therefore does not adversely affect decodeability in practice.

---

### Official Review · Reviewer_CoBp · 2025-10-31

**Soundness:** 2
**Presentation:** 1
**Contribution:** 2
**Rating:** 2
**Confidence:** 4

**Summary:**

This paper proposes a method for molecular property optimization, which operates within the latent space of a pre-trained Variational AutoEncoder (VAE). By employing the SELFIES representation, the VAE's latent space is reportedly free from the "dead zones" that typically plague models trained on traditional SMILES representations. This property is highly advantageous for optimization tasks. To render navigation of this high-dimensional latent space tractable, the authors propose learning property-specific transformations that project the latent vectors into a much lower-dimensional space. A covariance whitening regularizer is introduced to simplify the downstream optimization. Although the resulting optimization problem is non-convex, the authors employ a multi-restart strategy to mitigate the impact of local minima.

**Strengths:**

The central idea of performing property optimization within a learned, low-dimensional subspace—rather than in the VAE's full native latent space—is interesting. This approach to dimensionality reduction for targeted optimization represents the key contribution of the paper.

**Weaknesses:**

While the proposed low-dimensional projection is intriguing, the paper seems to concede (e.g., in the "Decoding Protocol" section) that the fundamental challenges of non-convexity and high-dimensionality persist.

A critical omission is a comparison against a baseline that predicts properties directly from the full VAE latent space (e.g., a simple regression model). Such a baseline would not only contextualize the benefit of the proposed projection but also highlight a significant weakness of the current method: its apparent inability to handle multi-property optimization, a capability that a direct-prediction model would inherently possess. The lack of multi-property optimization support is a major limitation.

Furthermore, a significant weakness of the paper is its lack of thorough contextualization. The "Related Work" section remains at a high level of abstraction, presenting lists of citations without specific discussion of how those works relate to the paper's novel contributions or limitations. Consequently, the paper fails to compare against any relevant, modern baselines. Even the VAE component's evaluation relies on a baseline from 2019, which is outdated given the rapid progress in the field.

To address this, the authors should position their work more clearly relative to:
- General structured prediction (have a look at papers like [1,2]).
- Modern representation learning for molecules, especially SELFIES-based models (e.g., [3] offers an encoder-decoder, [4] an encoder-only).
- Other contemporary methods for molecular property optimization.

[1] Amos, Brandon, Lei Xu, and J. Zico Kolter. "Input convex neural networks." International conference on machine learning. PMLR, 2017.
[2] LeCun, Yann, Chopra, Sumit, Hadsell, Raia, Ranzato, M,
and Huang, F. A tutorial on energy-based learning. Pre-
dicting structured data, 1:0, 2006.
[3] Priyadarsini, Indra, et al. "Self-bart: A transformer-based molecular representation model using selfies." arXiv preprint arXiv:2410.12348 (2024).
[4] Yüksel, Atakan, et al. "SELFormer: molecular representation learning via SELFIES language models." Machine Learning: Science and Technology 4.2 (2023): 025035.

# Typos:
1. L260: “itt” -> “it”

**Questions:**

1. Causal Decoder: The rationale for using a causal decoder is unclear. This architecture is typically employed for auto-regressive next-token prediction, which does not appear to be required by your non-autoregressive setup. How does the performance of this causal decoder compare to a standard (non-causal) self-attention mechanism?
2. Decoder Architecture: Could the authors please provide a diagram or pseudo-code for the decoder architecture? The description of how the property vector $m$ is repeated $T$ times and utilized within the model is currently difficult to follow.
3. Decoding Protocol Novelty: Is the "Decoding Protocol" a novel contribution of this work or standard practice? If it is novel, the justification for presenting it solely in the narrow context of molecular optimization, rather than as a general method for structured prediction, is missing. If it is standard practice, please provide relevant citations.
4. Tokenizer: What specific tokenizer is used for the SELFIES strings? Is it character-level, or does it utilize a specific SELFIES vocabulary?

---

> ### Author Response · Authors · 2025-11-26
> **Addressing Questions (Response to Reviewer CoBp) 1/3**
>
> We thank the reviewer for their time and feedback. We address the questions and weaknesses in detail below:
>
> **Questions**
>
> **1. Causal Decoder**
>
> The probability model described in the paper is explicitly autoregressive:
>
> $p(x_{1:T} \mid z) = \prod_{t=1}^{T} p(x_t \mid x_{<t}, z)$
>
> The causal mask is the critical input which enforces the correct conditional structure by ensuring that the prediction at position $t$ depends only on the preceeding tokens $x_{<t}$ (left-context) and the latent vector $z$ and never on the future ground-truth tokens.
>
> Concretely, the mask is a square matrix of shape $T \times T$ where $T$ is the target sequence length, and is a strictly lower-triangular matrix (including the diagonal) with entries indicating permissible attention (e.g. ones for allowed positions and zeros or
> −∞ for disallowed future positions). Without this masking, each token would have access to future ground-truth tokens during training, leading to information leakage and an invalid likelihood factorisation.
>
> It is important to note that the decoding is implemented in a single parallel pass (one-shot execution) hence the causal mask is necessary to ensure that the model respects the left-to-right generative semantics.
>
> Using a non-causal self-attention mechanism would allow each token to attend to the entire target sequence, including future positions, effectively changing the probabilistic interpretation of the decoder from an autoregressive to a fully conditional one. While such a formulation may be suitable for denoising-style or reconstruction-only objectives, it is inconsistent with the generative setting considered here, where the decoder must produce sequences sequentially at test time.
>
> So replacing the causal decoder with a non-causal self-attention variant will lead to extremely poor generation quality even if the losses at training time are comparable. This is expected, as the non-causal decoder learns to exploit future-token information unavailable at inference / generation time.
>
> We will revise the manuscript to make this motivation explicit. The decoder logic below will also help clarify this.
>
> **2. Decoder clarification and pseudo-code**
>
> z -> m (memory logic): We project the latent vector $z \in \mathbb{R}^{d}$ to a decoder embedding $m \in \mathbb{R}^{D}$ and tile it along the sequence dimension to form a “tiled memory” $M \in \mathbb{R}^{T \times D}$
> Conceptually, this tiling is needed as the entire target sequence is conditioned on the same global latent context derived from $z$ during training, this conditioning is applied in parallel across all positions. So each target token attends to the same latent vector $m$ at all decoder layers.
>
> Why not inject the raw latent $z$ directly? : this is mainly because the parameters to derive the memory context $m$ serves as an adapter to mould $z$ / latent information into a form that is relevant to prediction at each token step.
>
> To make the decoder logic extremely clear we provide the pseudocode below:
>
> The main ingredients to keep track of (during decoder training) are:
>
> - **Ground truth input sequence** (used for teacher forcing): $x_{\text{input}} = [sos, t1, t2, t3, ....]$ (without the eos token)
> - **Latent embedding:** $z \in \mathbb{R}^d$
> -  **Projection:** $m = W_z z + b_z \in \mathbb{R}^D$
> - **Tiled latent memory**$M = [m, m, \dots, m]^{T} \in \mathbb{R}^{T \times D} $
>
> At each decoder layer, there are only two critical steps happening:
>
> 1. Apply masked self-attention over token embeddings.
> 2. Apply cross-attention to the latent memory M.
>
> ```
> z → m = Linear(z)
> M = [m, m, ........m]   # repeat(m, T times)
>
> H^{(0)} = Embed(x_{input}) + EmbedPos(0:T-1)  # input embedding + position embedding
> H = H^{(0)}.   # H^{(0)} ∈ R^{T × D}
>
> for layer in DecoderLayers:
>     H_self = MaskedSelfAttention(H, H, H)                          # causal MSA (+ padding mask)
>     H_cross = CrossAttention(H_self, keys=M, values=M)  # latent conditioning at each time-step
>     H_ffn = FeedForward(H_cross)
> logits = Linear(H_ffn)
> ```
> This design allows the latent variable to exert global control across all positions, rather than injecting it only at initialisation, as is common in simpler VAE decoders.
>
> It is also important to disambiguate the subtle differences in the forward passes during training and generation.
>
> The training logic is basically:
>
> ```
> predict $x_{1}, x_{2}, ....x_{T}$ in one shot:
> with a mask on the sequence that ensures that only the left tokens ($x_{<t}$) are visible while processing $x_{t}$
> ```
> the generation logic is truly sequential as we don't have the ground truth:
> ```
> start with SOS -> predict x1-> append x1 -> predict x2 -> append x2 ......
> ```
> Both are autoregressive to the core (due to the causal mask) but the execution style during training is parallel and during generation is sequential.
> The probability model doesn't convey the execution logic and we agree, it needs to be a lot more detailed in the paper.

---

> > ### Author Response · Authors · 2025-11-26
> > **Addressing Questions and Weaknesses (Response to Reviewer CoBp) 2/3**
> >
> > **3. Decoding Protocol: Novelty and Scope**
> >
> > The proposed “Decoding Protocol” is a novel contribution of this work. It is not standard practice in VAEs or any other molecular generation framework. What we mean by the decoding protocol is the mechanism to decode the warped latent coordinates. This entails lifting them back to the original latent space which is compatible with the decoder.
> >
> > Specifically, the lifting procedure from warped coordinate $u_{j}^{*}$, back to the global latent coordinate $z$ via the optimisation (equation 13 in the paper) is given by the equation,
> >
> > $z^* = \arg\min_{z} \; \|T_j(z) - u_j^*\|_2^2 + \lambda \|z - z_0\|_2^2$
> >
> > This formulation explicitly handles the non-invertibility of the learned warping transformation $T_{j}$
> >  and provides a principled mechanism for decoding from the compressed, property-aligned subspaces.
> >
> > While demonstrated here in the context of molecular optimisation, this protocol is in fact general and applicable to any structured generation task where meaningful but non-invertible latent compressions are learned (e.g controllable text generation, disentangled image synthesis, conditional protein design). We will clarify this generality explicitly in the revised manuscript to avoid confusion.
> >
> > **4. Tokenizer for SELFIES**
> >
> > We use a vocabulary-based SELFIES tokenizer, not a character-level tokenizer.
> >
> > Specifically:
> >
> > - SELFIES strings are tokenised using the official `selfies` Python library.
> > - Each SELFIES token corresponds to a grammar unit such as [C], [Branch1], [Ring2], etc.
> > - These tokens are then integer-encoded and padded using the [nop] token to match the maximum sequence length in the dataset.
> > - This preserves semantic atomic units rather than degenerating into meaningless character-level fragments, a known limitation of SMILES processing, for instance SMILES would encode '(' and ')' as two characters whereas in SELFIES, the atomic unit is enveloped in the square brace.
> > An example of the integer encoding {[C] : 1, [O] : 2, [Ring2]: 3 ..... and so on}
> >
> > ## Weaknesses
> >
> > > A critical omission is a comparison against a baseline that predicts properties directly from the full VAE latent space (e.g., a simple regression model)
> >
> > Actually, the results of property prediction on a full VAE latent space (here, all properties are jointly modelled and predicted from the full latent space) are enclosed in the appendix 6.1 and Table 2. Despite, a compression factor of 64x (the full latent dim is 256 and warped coordinate space is just 4 dimensional), that is a 98.4% reduction in dimensionality we see no degradation in the prediction of properties (visually by the plots, Fig. 1 vs Appendix 6.1 or by the aggregated metrics in Table 2.)
> >
> > > The lack of multi-property optimization support is a major limitation.
> >
> > We respectfully disagree that the framework lacks support for multi-property optimisation. As described in Section 4 (Discussion, Limitations, Scope), the method deliberately learns independent, property-aligned warped subspaces for each molecular property, noting that there are “no practical limits to the number of property specific warped spaces that can be simultaneously learnt in this manner”. This architectural choice avoids the well-documented interference effects that arise when multiple competing properties are forced to share a single entangled latent space, which often leads to degraded reconstruction or weakened property control.
> >
> > The QED experiment in the paper focuses on single property optimisation but the extension to the optimisation objective is straightforward:
> >
> > Let's say that the objective was something like this: QED - logP
> >
> > The multi-objective optimisation problem can be written as,
> >
> > $  (u^\star_{\text{QED}}, u^\star_{\text{logP}})  = \arg\max_{u_{\text{QED}}, u_{\text{logP}}} (\alpha w_{\text{QED}}^\top u_{\text{QED}} - \beta w_{\text{logP}}^\top u_{\text{logP}} - \sum_{j \in \{\text{QED}, \text{logP}\}} \gamma_j \||u_j\||_2^2)$
> >
> > Since the linear heads are entirely property-specific, optimisation in each warped subspace remains decoupled. As a result, we retain the same closed-form ascent directions as in the single-property setting, with optimal trajectories aligned with each property vector
> > $w_{j}$, ensuring interpretable optimisation even under composite objectives. The section on optimisation in the paper remains valid and so does the math of ascent direction.
> >
> > (time permitting we will upload the revised manuscript demonstrating this experiment by the end of the discussion phase)

---

> > > ### Author Response · Authors · 2025-11-26
> > > **Addressing Questions and Weaknesses (Response to Reviewer CoBp) 3/3**
> > >
> > > > The "Related Work" section remains at a high level of abstraction, presenting lists of citations without specific discussion of how those works relate to the paper's novel contributions or limitations.
> > >
> > > We take note of this and will provide a more detailed and elaborated expansion with respect to the literature cited.
> > >
> > > >Consequently, the paper fails to compare against any relevant, modern baselines.
> > >
> > > We thank the reviewer for this important point and agree that clearer positioning relative to adjacent literature would strengthen the manuscript.
> > >
> > > The mentioned SELFIES-based models such as SELFormer and SELF-BART provide useful points of reference for SELFIE based models. However, it is important to note that they are molecular foundation models mainly aimed at representation learning so that their embeddings can be used downstream for various tasks.
> > >
> > > For instance, SELFormer is an encoder-only masked language model designed for molecular representation learning and downstream property prediction; it does not perform molecule generation or latent-space optimisation and therefore is not directly comparable as a generative or optimisation baseline. We will include this reference in related works.
> > >
> > > SELF-BART, by contrast, is an encoder–decoder model capable of unconditional molecular generation and reports strong performance on validity, novelty, and diversity; for a contextual comparison the UNV metrics for SELF-BART are 99.9%,100%, 99.8% and ours are 99.9%, 100%, 100%; and this is in-line with several other models. The more informative MC reconstruction accuracy is not directly available in the paper but we reckon can be computed. However, we need to keep in mind that  SELF-BART is trained on a very large, chemically diverse corpus (1B molecules across ZINC-22 + PubChem, ~354mn params). As such, comparisons on downstream MoleculeNet tasks or other predictive benchmarks are not like-for-like, given the significant differences in training regime and data scale.
> > >
> > > If our paper was proposing a new foundation model, it would be absolutely valid to take into account the recent foundation model papers on chemical sequence generation. As the focus of the paper in those cases is mainly to demonstrate the usefulness of embeddings in downstream evals. Our focus is entirely different. We are not proposing a new generative backbone or a foundation model. We are proposing a mathematical tehcnique to further compress the generative latent space to make it more conducive, controllable and interpretable for optimisation.
> > >
> > > The inclusion of metrics such as uniqueness, novelty, validity, and reconstruction accuracy in our paper is therefore intended primarily to demonstrate that the underlying generative architecture performs comparably to existing methods, ensuring that the proposed optimisation framework is not built on a weak generative base.
> > >
> > > Accordingly, we will expand the manuscript to include a discussion of recent foundation models such as SELFormer and SELF-BART and provide generation-quality metrics where available and expand Table 1 where we can. However, these models occupy a complementary rather than directly competitive methodological space. Importantly, our framework is fully compatible with such models: the proposed warping and optimisation technique could, in principle, be applied to latent spaces learned by generative chemical foundation models. That said, property alignment at such scales may not always be feasible due to the lack of property annotations for extremely large chemical corpora, which would necessitate modifications to the alignment loss formulation.
> > >
> > > **Request**
> > >
> > > We sincerely hope the reviewers will take cognizance of the responses provided above, as we believe we address many of the points of concern. And if these responses were useful, we respectfully request that the reviewers consider revisiting and updating their scores within the discussion phase. Given, the several challenges with the review process this year for many papers, we sincerely hope the reviewers would provide a responsible assessment.

---

### Official Review · Reviewer_fxRp · 2025-11-01

**Soundness:** 2
**Presentation:** 2
**Contribution:** 1
**Rating:** 0
**Confidence:** 4

**Summary:**

This paper presents a VAE framework to learn property-aligned subspace via an auxiliary loss. Results show that the proposed method can learn a subspace where molecules with similar property values have similar latent representations.

**Strengths:**

The presentation is generally clear.

**Weaknesses:**

- Generally, this paper introduces nothing new to the community. The overall problem, using VAE latent space traversal for property optimization, is something that this community was exploring four or five years ago. Many more effective optimization methods have been proposed over the past five years; unfortunately, none of them are compared with or even mentioned in this paper.
- Even for vanilla molecule generation without property optimization, all the comparison methods in Table 1 are from 2020 or earlier, despite the rapid progress in this field over recent years. The lack of up-to-date baselines makes the evaluation unconvincing.
- The idea of using a Transformer-based VAE is also not new. Simply replacing SMILES with SELFIES does not contribute meaningful novelty either. The only new component, the latent space contrastive loss (Eq. 6), still does not address a valid problem. In the introduction, the authors criticize latent space disentanglement methods for suffering from property entanglement, but this issue has already been studied and mitigated by a series of works addressing latent factor correlation in VAEs.
- The properties used for evaluation, including QED and LogP, are considered toy properties with limited connection to real-world molecular design tasks, which further weakens the practical significance of the reported results.

**Questions:**

I don't have more questions.

---

> ### Author Response · Authors · 2025-11-27
> **Response to Reviewer fxRp (1/3)**
>
> We thank the reviewe for their time.
>
> **Misunderstanding key methodological point**: First, we would like to clarify that while we use the term “subspace” for intuition, we clarify multiple times in the manuscript that the warped representation is not a subspace of the original latent space (it is not a plane carved out of the original latent embedding). Instead, the transformation $T_{j}: \mathbb{R}^{d} \longrightarrow \mathbb{R}^{k}$ defines a newly learned coordinate space with its own geometry. This transformation constructs a property-aligned coordinate system rather than extracting a subset of existing dimensions. This is obvious as we learn the new $k-$dimensional coordinate space (per property) after training the generative backbone. It wouldn't make sense at all to go around changing coordinates in subspaces of the original latent space ..distorting the information in the latents.
>
> We will clarify this in a much more direct way in the manuscript.
>
> We respectfully disagree with the reviewer’s assertion that this work “introduces nothing new” or fails to engage with the current state of the field.
>
> **1. “VAE latent space traversal for property optimisation is old.”**
>
> Our work does not perform traversal or optimisation in the original VAE latent space. We explicitly depart from classical latent-space walking, gradient ascent, or interpolation strategies that operate directly on the entangled high-dimensional latent vectors learned by a VAE. Instead, we learn a distinct, lower-dimensional warped coordinate space through a dedicated transformation $T_{j}: \mathbb{R}^{d} \longrightarrow \mathbb{R}^{k}$, constructed with a specific inductive bias that aligns geometry with molecular property variation. In contrast to prior “latent traversal” methods, which operate over an unstructured and globally entangled latent embedding, our approach introduces a new geometry whose structure is explicitly shaped by the alignment objective and covariance regularisation. The VAE latent space is not the optimisation domain.
>
> **2. “Transformer-VAE with SELFIES is not novel.”**
>
> Correct. We already say **clearly** that many papers in literature have used different formulations of transformer vaes. We only recapitulate state-of-the-art performance for our version of the model to establish it as a competitive back-bone on which the other aspects of the model and its extensions are built. It's a baseline we need to include the paper, and Table 1 confirms that we achieve SOTA performance of UNV metrics (uniqeness, novelty, validity). As well as SOTA performance in MC reconstruction (% fraction of molecules reconstructed exactly when stochastically encoded and decoded multiple times).
>
> However, the MC reconstruction accuracy is only available in a few papers and lot of recent work does not report this metric. Hence, **we chose to report it along with the papers which do in fact report this metric**. in recent years there seems to be a broad consensus  that reconstruction quality / accuracy is not a  strong indicator of generative capabilities in terms of sampling and diversity..and can in some cases reflect overfitting or an overly powerful decoder which can detract from sampling ability. Nevertheless, we report both standard and Monte Carlo reconstruction accuracy as a diagnostic of latent bottleneck fidelity and decoding stability, particularly since reliable reconstruction is important for interpreting traversal behaviour in learned coordinate spaces we create.
>
> To the best of our knowledge, and from compiling all the papers below (2021-2024) the MC reconstruction for ZINC 250 has not been reported for SELFIES with a transformerVAE generative model.
>
> **3. “Latent contrastive loss doesn’t solve a valid problem.”**
>
> This criticism stems from a misunderstanding of both the failure mode we target and the role of the proposed loss. The problem we address is not generic “disentanglement” in the sense of encouraging statistically independent latent dimensions, but rather the practical and well-documented issue that standard VAE latent spaces exhibit geometry that is poorly aligned with property variation. In such spaces, gradients for property optimisation are often dispersed across entangled and semantically irrelevant directions, producing erratic or non-monotonic changes in the decoded molecules.
>
> Our objective explicitly targets this geometric misalignment by learning a warped latent representation where Euclidean distance in this warped coordinate system reflects differences in molecular properties, thereby enforcing smooth, directional traversal. This alignment between distance and property variation is not addressed by existing disentanglement methods, which focus on factor independence or decorrelation but do not ensure that movement in latent space corresponds to predictable changes in specific molecular properties. The problem we solve, therefore, is both valid and practically motivated.
>
> (continued below)

---

> ### Author Response · Authors · 2025-11-27
> **Response to Reviewer fxRp (2/3)**
>
> Secondly, while the loss shares superficial similarity with contrastive formulations in that it operates on pairwise samples, **it is not a standard contrastive loss**. Classical contrastive objectives construct positive and negative pairs and aim to collapse similar items. In contrast, our alignment loss is metric-preserving and regression-driven: it enforces a proportional relationship between pairwise distances in the warped coordinate space and pairwise differences in continuous property values. In essence, our objective is better understood as a **distance alignment or geometric regularisation loss**.
>
> **4. “QED and LogP are toy properties.”**
>
> QED and logP are simplified proxy properties and are not intended to represent the full complexity of real-world drug discovery objectives. However, the reviewer’s criticism overlooks the fact that these metrics remain the **de facto standard benchmarks across the molecular generative modelling literature, including in the most recent VAE-based and non-VAE approaches**. Their widespread use is not arbitrary: they are continuous, reproducible, and widely available at scale, making them uniquely suited for  evaluation of generative and optimisation behaviour. for instance, we showed in our QED experiment that the model can propose molecules with SOTA QED scores (as reported on ZINC 250).
>
> The purpose of using QED, SAS and logP is not clinical relevance but providing a testbed to demonstrated a mathematical tehcnique of geometric regularisation. **Researchers who have access to expensive to compute biological properties can very well adapt the technique to their own properties but we are looking at it purely from a methodological perspective not a clinal one**. Nothing in the proposed framework is tied specifically to QED or logP. The method is agnostic to the choice of property and is, in fact, designed to operate on any scalar-valued signal, we are only using what is a community accepted benchmark for demonstrating chemical generative techniques. I'm sure if we didn't include it, it would also invite a critique.
>
> The point is that the properties are widely available proxies used in almost every reference on this topic. The aim is to propose a general technique that can potentially even be demonstrated on other types of datasets outisde of chemistry like: images, material design, biological sequences where the supervisory signal can be any continuous scalar value. For example, in an image dataset annotated with a continuous sharpness or blur score, the proposed warping mechanism would learn a coordinate space in which images of comparable focus quality are embedded nearby. And we can uncover something like a 'blur-axis' decoding along which should give use images of increasing sharpness.
>
> **5. “More effective optimisation methods exist and are not compared / Baselines are outdated”**
>
> To address this we include the table below where we have listed several approaches which use a VAE style architecture on string sequences of molecules. UNV stands for uniqueness, novelty, validity.
>
> | Full citation | Model name | UNV reported? | Molecular representation |
> |--------------|------------|----------------|---------------------------|
> | Iwata, H. et al. (2023). *VGAE-MCTS: A New Molecular Generative Model Combining the Variational Graph Auto-Encoder and Monte Carlo Tree Search.* Computational Chemistry. | VGAE-MCTS | Yes | Graphs |
> | Liu, Q. et al. (2022). *SELF-BART: A Transformer-based Molecular Representation Model using SELFIES.* arXiv preprint. | SELF-BART | Yes | SELFIES |
> | Yüksel, A. et al. (2023). *SELFormer: Molecular representation learning via SELFIES language models.* Machine Learning: Science and Technology, 4(2). | SELFormer | Yes | SELFIES |
> | Seo, S. et al. (2023). *Molecular Generative Model via Retrosynthetically Prepared Chemical Building Block Assembly.* Advanced Science. | Building Block Assembly Model | Yes | Graphs / Building-block graphs |
> | Dollar, O. et al. (2021). *Attention-based generative models for de novo molecular design.* Chemical Science. | Attention-based Generative Model | Yes | SMILES |
> | Fan, X. et al. (2023). *ICVAE: Interpretable Conditional Variational Autoencoder for De Novo Molecular Design.* Journal of Chemical Information and Modeling. | ICVAE | Yes | SMILES |
> | Ochiai, T. et al. (2023). *Variational autoencoder-based chemical latent space for large molecular structures with 3D complexity.* Communications Chemistry, 6, 249. | NP-VAE | Yes | SMILES (3D-aware encoding) |
>
> They all report SOTA 99.9% - 100% UNV  metrics so that is largely a solved constraint. Further, they report QED optimisation in the latent space, also largely SOTA numbers qed > 0.92, or report in terms of average across a validation set.

---

> ### Author Response · Authors · 2025-11-27
> **Response to Reviewer fxRp (3/3)**
>
> Our model reports the same SOTA UNV metrics as well as close to SOTA QED optimisation but the main point worth noting, is that we are not doing this opitmisation through a traversal in the canonical high-dimensional latent space but a low-dimensional learnt coordinate space.
>
> (However, we will update Table 1 in the paper to include these newer references, fyi none of these papers mention a comparable reconstruction accuracy for ZINC250k except for NP-VAE which reports it on larger drug compounds, 81% and the Attn VAE paper from the table reports 99.8% accuracy for SMILES (on ZINC250) but much lower validity of 56%)
>
> **The central contribution of this work does not lie in marginal improvements to these already saturated metrics**. Rather, the key point is **how this optimisation is achieved**. Unlike conventional approaches, which perform property optimisation through traversal or gradient-based manipulation within the canonical high-dimensional VAE latent space, our method explicitly decouples optimisation from this entangled representation. Instead, optimisation is carried out in a low-dimensional, learned coordinate space that is constructed with a specific inductive bias.
>
> **An important consequence of this is that it yields a representation in which the target property can be accurately predicted using a simple linear head** (page 5). This indicates that property variation has been effectively linearised and disentangled within the learned coordinate space.
>
> To the best of our knowlege we haven't seen a similar mathematicla technique used in generative modelling literature before. Agreed that we have subspace methods but this is not exactly a subspace method.
>
> **Request**
>
> We would we respectfully request that the reviewer reconsider their assessment of the manuscript. Several of the concerns raised stem from not fully taking into account the core methodology and scope of the contribution, particularly regarding the nature of the learned warped coordinate space, the role of the alignment regularisation, and the objectives of the evaluation framework. We believe that the manuscript presents a meaningful methodological advance in how property-aligned geometry can be learnt in any setting, not just chemisty; and this contribution is not adequately reflected in the current score.

---

### Meta-Review · Area_Chair_hDC4 · 2026-01-06

**Summary:**

The reviews vary substantially in their level of detail and technical engagement. Some feedback lacks sufficient justification for the assigned scores, providing limited analysis of the paper’s methodology, theoretical formulation, or experimental results. As a result, not all evaluations can be weighted equally in the final decision.

Independently of this issue, the paper has several substantive weaknesses that affect its overall strength. The motivation is not presented clearly, making it difficult to understand the necessity of the proposed approach relative to existing methods. The algorithmic design appears overly complex, with key design choices insufficiently explained or justified. The writing and organization further reduce clarity, obscuring the main contributions. In addition, parts of the experimental evaluation seem insufficiently motivated and do not clearly support the paper’s core claims.

Overall, while some reviews lack depth, the paper itself would benefit from significant improvements in motivation, clarity of exposition, and experimental focus before it can be considered competitive for acceptance.

**Reviewer Concerns:**

To the reviewer CoBp's concerns:

A key concern is the lack of appropriate baselines, which limits the ability to assess the practical significance of the proposed method. In addition, several sections of the paper appear misaligned with the main focus and may be misleading with respect to the central contributions; these issues were not adequately addressed during the rebuttal.

More broadly, the paper lacks thorough contextualization. Different components of the work are not clearly positioned relative to the stated goals, leading to confusion about their relevance. For example, Sections 2.1 and 3.1 have a weak connection to the paper’s title and core narrative, which undermines the coherence of the overall presentation.

**Reviewer Scores:**

According to their evaluations, all of the reviewers may keep the scores.

---

### Decision · Program_Chairs · 2026-01-26

Reject